# Anatomical and single-cell transcriptional profiling of the murine habenular complex

Michael L Wallace*, Kee Wui Huang, Daniel Hochbaum, Minsuk Hyun, Gianna Radeljic, Bernardo L Sabatini*

Department of Neurobiology, Howard Hughes Medical Institute, Harvard Medical School, Boston, United States

**Abstract** The lateral habenula (LHb) is an epithalamic brain structure critical for processing and adapting to negative action outcomes. However, despite the importance of LHb to behavior and the clear anatomical and molecular diversity of LHb neurons, the neuron types of the habenula remain unknown. Here, we use high-throughput single-cell transcriptional profiling, monosynaptic retrograde tracing, and multiplexed FISH to characterize the cells of the mouse habenula. We find five subtypes of neurons in the medial habenula (MHb) that are organized into anatomical subregions. In the LHb, we describe four neuronal subtypes and show that they differentially target dopaminergic and GABAergic cells in the ventral tegmental area (VTA). These data provide a valuable resource for future study of habenular function and dysfunction and demonstrate neuronal subtype specificity in the LHb-VTA circuit.

## Introduction

The habenula is an epithalamic structure divided into medial (MHb) and lateral (LHb) subregions that receives diverse input from the basal ganglia, frontal cortex, basal forebrain, hypothalamus and other regions involved in processing both sensory information and internal state (*Herkenham and Nauta, 1977*; *Yetnikoff et al., 2015*). Two main targets of the LHb output are major monoaminergic structures in the brain, the ventral tegmental area (VTA) and raphe nuclei (dorsal (DRN) and medial (MRN)), whereas the MHb targets the interpeduncular nucleus (*Herkenham and Nauta, 1979*). Due to its dual effects on dopamine and serotonin-producing neurons, LHb has been proposed to contribute to the neurobiological underpinnings of depression and addiction (*Li et al., 2011*; *Li et al., 2013*; *Maroteaux and Mameli, 2012*; *Meye et al., 2016*). Furthermore, the LHb has been implicated in a wide range of functions and behaviors including reward prediction error, aversion, cognition, and adaptive decision making (*Hikosaka, 2010*; *Matsumoto and Hikosaka, 2007*; *Mizumori and Baker, 2017*; *Proulx et al., 2014*; *Tian and Uchida, 2015*; *Wang et al., 2017*).

The effects of LHb on each downstream structure and its contribution to different behaviors are likely carried out by distinct populations of neurons. In addition, the LHb may, at the macroscopic level, consist of distinct sub-domains with differential contributions to limbic and motor functions (*Zahm and Root, 2017*). Further, many studies suggest that habenular neurons show differences in gene expression across hemispheres (*Concha and Wilson, 2001*; *Pandey et al., 2018*), projection targets (*Quina et al., 2015*), and anatomical location (*Gonçalves et al., 2012*). Nevertheless, the systematic relationship between molecular profiles, projections patterns, and anatomical organization of neurons in the LHb is unknown.

Here, we provide a comprehensive description of the neuronal subtypes in the LHb based on single-cell transcriptional profiling, multiplexed fluorescent in situ hybridization (FISH), and cell-type-specific monosynaptic retrograde tracing. Furthermore, as the MHb was included in our dissections, we also provide a molecular description of this nucleus. We find that the MHb has five, and LHb has four, transcriptionally defined neuronal subtypes. Interestingly, the HbX subtype, which lies at the

*For correspondence:
mwallace1984@gmail.com (MLW);
bsabatini@hms.harvard.edu (BLS)

Competing interests: The authors declare that no competing interests exist.

border between the MHb and LHb, is more transcriptionally similar to other LHb subtypes than MHb subtypes. We find that the four LHb neuronal subtypes are distinct and monosynaptic retrograde tracing revealed that they differentially target the dopaminergic and GABAergic neurons of the VTA, as well as the dorsal raphe nucleus. Furthermore, we find the LHb is organized into subregions defined by these transcriptionally discriminable neuronal subtypes. Together we identify previously unknown neuronal heterogeneity in the habenula and reveal that different neuronal classes are biased in their projection targets.

## Results

### Cell type composition of the habenula by transcriptomic profiling

To examine cellular heterogeneity in the habenula, we performed high-throughput single-cell transcriptional profiling ('InDrop') (*Klein et al., 2015*). Cell suspensions from the habenula were generated from acute, microdissected brain slices from adult mice (*Figure 1A*), producing 25,289 single-cell transcriptomes (SCTs). Excluding SCTs with <200 genes,<500 UMIs, or >10% mitochondrial genes resulted in 7,506 SCTs that were further analyzed. This subset had median counts of 2593 UMIs (min = 501, max = 17787, IQR = 3986) and 349 genes (min = 302, max = 5276, IQR = 1733) per cell (*Figure 1—figure supplement 1C*). Subsequent analysis of gene expression patterns by principal components (PC) analysis and shared-nearest-neighbors (*Satija et al., 2015*) resulted in 12 cellular clusters (*Figure 1B*, see Materials and methods for details on sequential clustering). Major cell classes (i.e. neurons, astrocytes, microglia, etc...) within these clusters were identified by expression of cell-type-specific gene combinations that were extensive cross-referenced with published datasets (*Saunders et al., 2018*; *Zeisel et al., 2018*) (*Figure 1B–C*). Our data confirmed recent work on non-neuronal cell-types in habenula describing high levels of *Tnf* in microphages and microglia (*Valentinova et al., 2019*) and high levels of Kir4.1 (*Kcnj10*) in astrocytes (*Cui et al., 2018*) (*Figure 1—figure supplement 2*). In contrast to other species (*Pandey et al., 2018*), no major transcriptional differences were observed (*Figure 1—figure supplement 1A–B*) across left and right hemispheres; therefore, cells from both hemispheres were pooled for analysis. The majority of the cells in the dataset were neurons (53%) and we focused our analysis on these clusters for the remainder of the study.

Neurons (*n* = 3,930 cells), identified by expression of genes required for chemical synaptic transmission such as *Snap25*, *Syp,* and *Syt4,* clustered into two main classes (*Figure 1B–C*). We examined if these two neuronal clusters could be spatially distinguished using digital in situ hybridization (ISH) analysis (Allen Brain Atlas, [*Lein et al., 2007*]) of differentially expressed genes (*Finak et al., 2015*). The larger cluster of neurons (n = 3,370 cells) expressed *Tac2* and corresponds to the MHb (*Figure 2*), whereas the smaller cluster (n = 560 cells) expressed *Gap43* and corresponds to the LHb (*Figure 3*).

### Differential gene expression reveals the spatial organization of MHb neuron subtypes

Analysis of MHb neurons revealed that they could be divided into eight clusters (*Figure 2—figure supplement 1A*). However, three clusters were clearly distinguished by high expression of activity-dependent genes (ADGs) (*Figure 2—figure supplement 1B*), suggesting that they might simply reflect neurons of other clusters that had been recently strongly activated. Indeed, regressing out the PC containing a large number of ADGs (*Figure 2—figure supplement 1E–F*) caused these three high ADGs clusters to merge with other MHb clusters (*Figure 2—figure supplement 1C–D*), leaving five distinct subtypes of MHb neurons.

We constructed a cluster dendrogram using the averaged cluster gene expression to examine the transcriptional differences between these subtypes (*Figure 2D*). In general, subtypes of MHb neurons were divided by genes that were involved in the synthesis and packaging of different neurotransmitters and neuropeptides. All MHb neurons expressed high levels of *Slc17a6* and *Slc17a7*, the genes encoding vesicular glutamate transporters 1 and 2, and *Tac2* suggesting that all MHb neurons are glutamatergic and produce the neuropeptide Neurokinin B (*Figure 2D*, *Figure 2—figure supplement 3*). Two of the five clusters also expressed *Slc18a3* and *Chat* (not shown), the vesicular transporter and biosynthetic enzyme for acetylcholine, respectively, indicating that these neurons

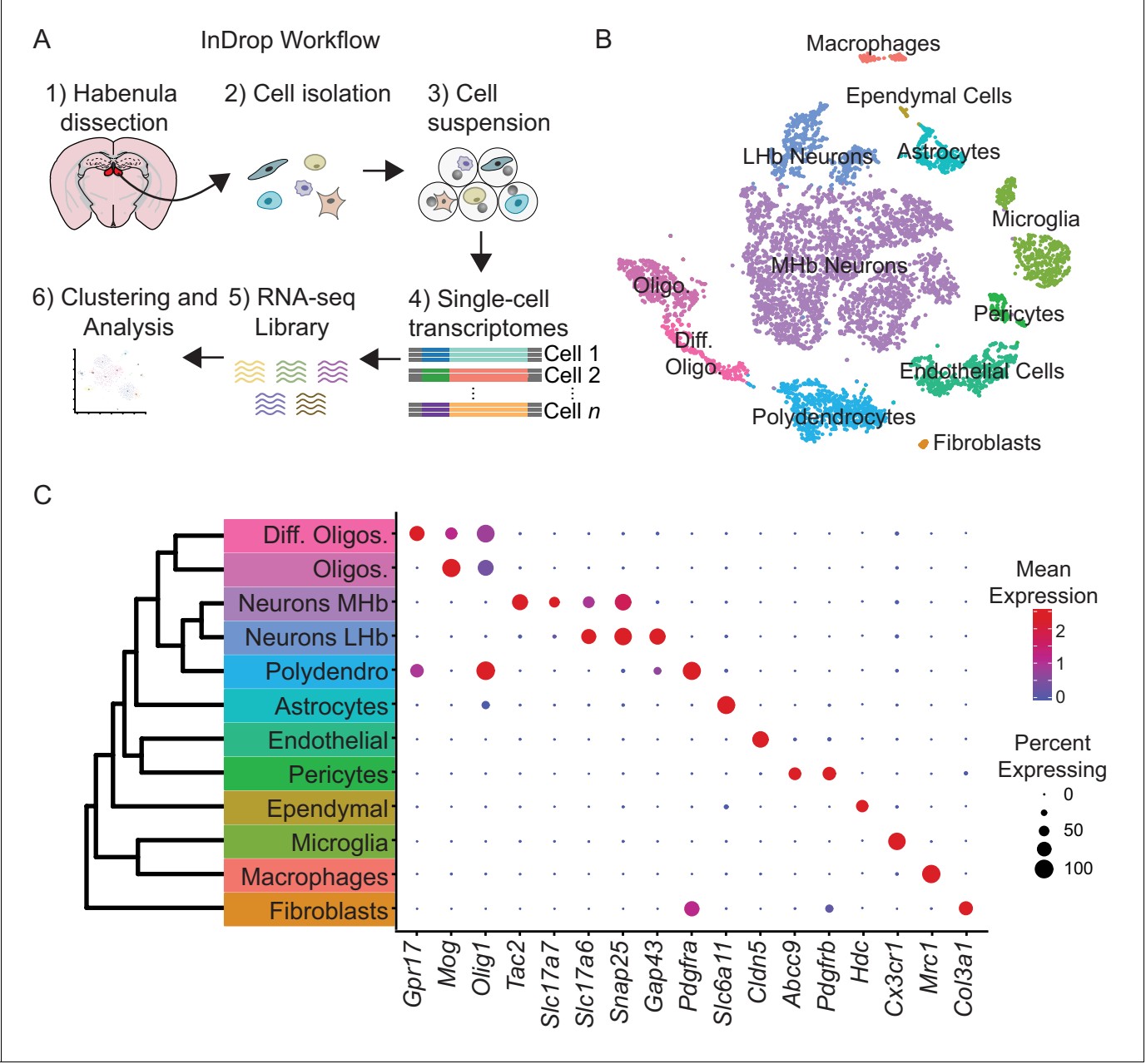

**Figure 1.** High-throughput single-cell transcriptomic profiling of the habenula. (**A**) Schematic for scRNA-seq using the inDrop platform. Tissue containing the habenula was microdissected from acute coronal brain slices prepared from adult mice (1). Tissue chunks were digested in a cocktail of proteases and followed by trituration and filtration to obtain a cell suspension (2). Single cells were encapsulated using a droplet-based microfluidic device (3) for cell barcoding and mRNA capture (4). RNA sequencing (5) and bioinformatics analysis followed (6). (**B**) t-SNE plot of the processed dataset containing 7,506 cells from six animals. Cells are color-coded according to the cluster labels shown in (**C**). (**C**) Left: Dendrogram with cell class labels corresponding to clusters shown in (**B**). Right: Dot plot displaying expression of example enriched genes used to identify each major cell class. The color of each dot (blue to red) indicates the relative log-scaled expression of each gene, whereas the dot size indicates the fraction of cells expressing the gene.

The online version of this article includes the following figure supplement(s) for figure 1:

**Figure supplement 1.** Comparison of cell-type composition across hemispheres and gene diversity, mitochondrial genes, and UMIs across cell types.
**Figure supplement 2.** Expression of genes known to be important for habenular microglial and astrocytic function.

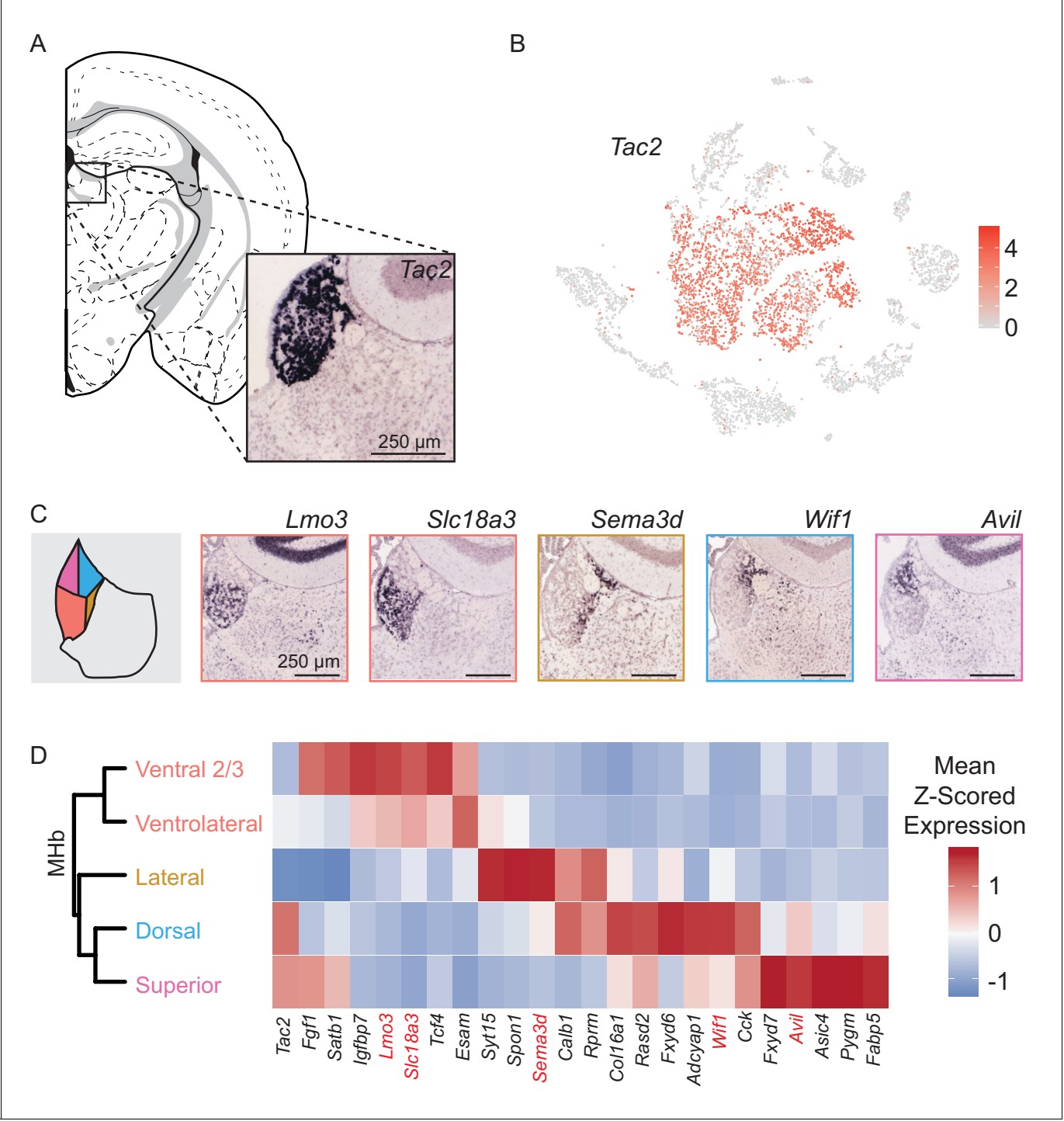

**Figure 2.** MHb neuron subtypes can be distinguished transcriptionally. (A) Location of MHb and ISH of *Tac2* expression from the Allen Institute Database. *Tac2* expression is restricted to cells in the MHb in this region. (B) *Tac2* serves as an excellent marker for MHb neurons in the dataset of SCTs (Scale on right shows normalized (log) gene expression.) (C) Left: Illustration showing patterns of gene expression observed for DEGs using the Allen Institute Database. Right: Sample ISH images from the Allen Institute Database showing selected differentially expressed genes for distinct transcriptionally defined neuronal subtypes in MHb. (D) Left: Dendrogram with MHb subtype labels corresponding to clusters shown in (*Figure 2— figure supplement 1C*). Right: Heatmap showing the relative expression (mean z- scored) of selected genes that are enriched in each MHb neuron subtype. Spatial distributions of enriched genes highlighted in (C) are labeled in red.

The online version of this article includes the following figure supplement(s) for figure 2:

*Figure 2 continued on next page*

*Figure 2 continued*

**Figure supplement 1.** Subclustering of MHb neurons before and after subtraction of heterogeneity introduced by elevated expression of activity-dependent genes (ADGs).

**Figure supplement 2.** Sample ISH images showing spatial distribution of selected differentially expressed genes in MHb.

**Figure supplement 3.** Differentially expressed genes define distinct habenular subtypes.

**Figure supplement 4.** Ion channel diversity in MHb and LHb neuronal subtypes.

**Figure supplement 5.** Ion channel diversity in MHb and LHb neuronal subtypes (part 2).

**Figure supplement 6.** Neurotransmitter receptor diversity in MHb and LHb neuronal subtypes.

**Figure supplement 7.** Neurotransmitter receptor diversity in MHb and LHb neuronal subtypes (part 2).

**Figure supplement 8.** GPCR diversity in MHb and LHb neuronal subtypes.

**Figure supplement 9.** GPCR diversity in MHb and LHb neuronal subtypes (part 2).

**Figure supplement 10.** GPCR diversity in MHb and LHb neuronal subtypes (part 3).

**Figure supplement 11.** GPCR diversity in MHb and LHb neuronal subtypes (part 4).

may co-release glutamate and acetylcholine (*Figure 2—figure supplement 3*), as has been described in several studies (*Ren et al., 2011*; *Soria-Gómez et al., 2015*). Interestingly, no MHb neurons expressed significant levels of the canonical GABA handling genes *Slc32a1*, *Gad1*, or *Slc18a2* (although *Gad2* was expressed at low levels in all subtypes); therefore, they are unlikely to release GABA.

To examine the spatial distribution of MHb neuron subtypes we cross referenced their differentially expressed genes (DEGs) with the Allen Mouse Brain Atlas of ISH hybridization data (*Supplementary file 3*) (*Lein et al., 2007*). Generally, we found that individual DEGs for particular MHb subtypes consistently mapped onto discrete regions in the MHb (*Figure 2C*, *Figure 2—figure supplement 2*). Also, DEGs for MHb neurons were rarely DEGs for LHb neurons (*Figure 2—figure supplement 3*). This permitted classification of transcriptionally defined MHb subtypes to particular subregions of MHb (*Figure 2D*, *Figure 4—figure supplement 1*). MHb neurons divided along the dorsal/ventral axis with a third lateral (enriched for genes *Sema3d*, *Calb1*, and *Spon1*) subtype (*Figure 2C–D*, *Figure 2—figure supplement 2C–E*). Ventral groups could be further subdivided into two distinct subtypes, the 'ventral two thirds' of the MHb (enriched for *Lmo3*) and the 'ventrolateral' MHb (enriched for genes *Esam* and *Slc18a3*). Gene expression patterns indicated that it is possible that neurons from these two subtypes were partially intermingled and did not form a defined border (*Figure 2C*, *Figure 2—figure supplement 2A–B*). The rest of the MHb could be subdivided into the 'dorsal' (enriched for genes *Col16a1*, *Wif1*, and *Adcyap1*) and 'superior' (enriched for genes *Cck* and *Avil*) subtypes. These two groups split along a medial/lateral axis with the 'dorsal' being more laterally located than the 'superior' (*Figure 2C–D*, *Figure 2—figure supplement 2F–I*) (*Wagner et al., 2016*; *Wagner et al., 2014*).

## Genetic distinction of four LHb neuron subtypes

*Gap43* is highly expressed in the LHb and along with several other genes (*Htr2c*, *Pcdh10*, *Gabra1*, and *Syn2*) distinguishes neurons of this region from those of neighboring MHb (*Figure 3A–B*, *Figure 2—figure supplement 3*). Unlike for MHb neurons, we did not detect significant elevation of ADGs in LHb neurons. We found four distinct clusters of neurons in LHb which, again unlike MHb, did not have distinct expression profiles of genes involved in the synthesis and packaging of different typical fast neurotransmitters (e.g. glutamate, GABA, acetylcholine) – all LHb neurons expressed high levels of *Slc17a6* and very low levels of *Slc32a1*, suggesting that they are glutamatergic.

Subdivisions of the lateral habenula based on topographic, morphological and cytochemical criteria have been described in rat (*Andres et al., 1999*) and mouse (*Wagner et al., 2016*; *Wagner et al., 2014*) and are used here to describe the patterns of DEGs extracted from our single-cell dataset (see terms in quotes below). We examined the spatial distribution of LHb neuron subtypes by cross referencing their DEGs with the Allen Mouse Brain Atlas of ISH data (*Lein et al., 2007*). We found that DEGs showed four distinct, but consistent patterns that aligned with their subclusters (*Figure 3C–D*). These consisted of 1) a cluster that showed high expression of DEGs in both the 'lateral oval' and 'central medial' subdivision, we named this the *oval/medial* subdivision; 2) a cluster that showed high expression of DEGs in the 'marginal subdivision of the medial division of the LHb', we called this the *marginal* subdivision; 3) a cluster that showed high expression of DEGs

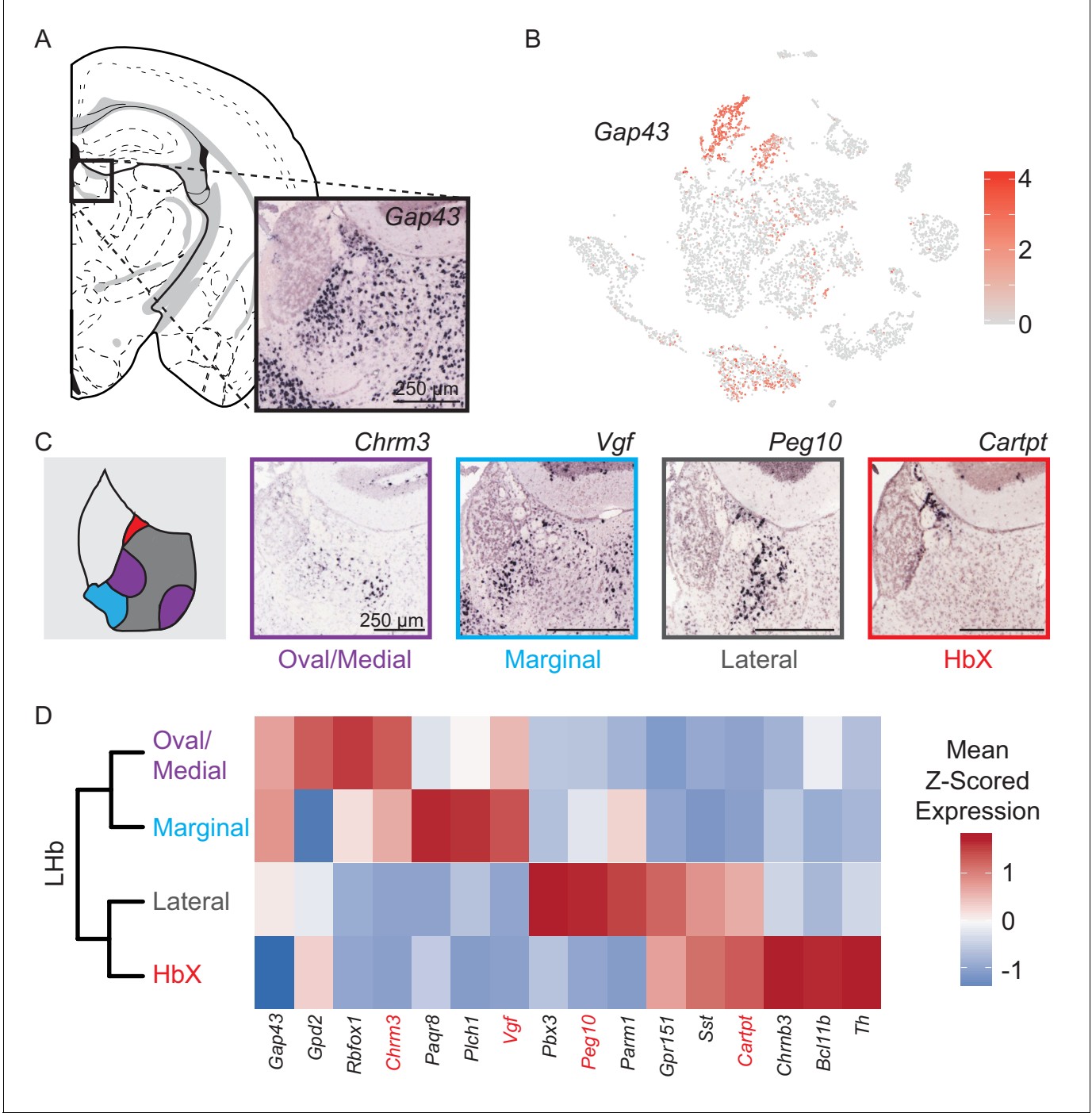

**Figure 3.** Characterization of genes differentially expressed between LHb neuron subtypes. (**A**) Location of LHb clusters and ISH *Gap43* expression from the Allen Institute Database. *Gap43* is highly expressed in neurons of the LHb and surrounding thalamus in this region, but excluded from MHb neurons. (**B**) *Gap43* serves as an excellent marker for LHb neurons in the dataset of single-cell transcriptomes (Scale on right shows normalized (log) gene expression.) (**C**) Left: Illustration showing patterns of gene expression observed for DEGs. Right: Sample ISH images from the Allen Institute Database showing selected differentially expressed genes for distinct transcriptionally defined neuronal subtypes in LHb. (**D**) Left: Dendrogram with LHb neuron labels corresponding spatial locations of differentially expressed genes within the LHb. Right: Heatmap showing the relative expression of selected genes that are enriched in each LHb neuron subtype. Spatial distributions of enriched genes highlighted in (**C**) are labeled in red.

in the 'lateral' subdivision (but avoiding expression in the 'lateral oval'), we also called this the *lateral* subdivision; and 4) a cluster that showed high expression of DEGs in the subdivision defined as 'HbX' lying on the dorsal border between MHb and LHb, we also refer to this as the *HbX* subdivision (*Figure 3C*) (*Wagner et al., 2016*; *Wagner et al., 2014*). Interestingly, the *HbX* region is more closely related in its gene expression to other LHb clusters than to any clusters in the MHb; therefore, it is more similar to LHb neurons than previously recognized (*Figure 3D*) (*Wagner et al., 2016*).

We performed multiplexed FISH to confirm the four transcriptionally defined clusters of LHb neurons were distinct and to further examine their spatial organization within the LHb. We chose four highly expressed DEGs that represented each LHb cluster (*Chrm3, Vgf, Gpr151*, and *Sst*) and examined their expression in individual neurons (*Figure 4*). As predicted by the single-cell sequencing, the chosen genes generally expressed in different cells, confirming that they defined molecularly distinct neuronal subtypes (*Figure 4*). An exception to this general rule, but consistent with the predictions of single-cell sequencing, individual neurons in the HbX expressed both *Sst* and *Gpr151* (*Figure 4D*). Additionally, when strongly expressed, *Chrm3* and *Vgf* were found in different cells, but they were occasionally co-expressed in neurons that had relatively low levels of both genes (*Figure 4A*).

The chosen genes are largely expressed in non-overlapping patterns at the macroscopic level, confirming the organization of LHb into molecularly defined subregions (*Figure 4*, *Figure 4—figure supplement 1*). More specifically, cells expressing DEGs from different subtypes occupied distinct spatial domains along the medial/lateral axis of the LHb (*Figure 4—figure supplement 2*). However, cells from a subtype did intermingle with cells of another group and sharply defined borders between LHb subregions were not observed (e.g. *Figure 4C*). Therefore, diagrams of gene expression (*Figure 4—figure supplement 1*) illustrate where gene expression is greatest or where cells expressing the gene are most numerous and not that gene expression is perfectly restricted to a particular subregion. Furthermore, heterogeneity of gene expression across subregions was most pronounced in the middle of the anterior/posterior axis of the habenula (see *Figure 4—figure supplement 1* and Methods).

To confirm that the gene expression patterns described above were representative of a subgroup rather than specific to those particular DEGs, we chose three additional DEGs form the oval/medial, marginal, and lateral subdivisions of the LHb (*Rbfox1, Plch1* and *Pbx3*, respectively, *Figure 3D*). Multichannel FISH for this second set of genes combined with the previous set confirmed that expression of DEGs from the same subgroup overlap both regionally, and in single cells (*Figure 4—figure supplements 2* and *3*). Importantly, we did not observe obvious gradients in expression along the medial/lateral axis for most of the DEGs using FISH; however, *Plch1* expression did decrease gradually for cells positioned more lateral within the LHb (*Figure 4—figure supplement 2E*).

## LHb neuron subtypes differentially target VTA GABAergic and dopaminergic neurons

The LHb projects via the fasciculus retroflexus to the ventral tegmental area (VTA), rostromedial tegmental area (RMTg), and median/dorsal raphe (*Herkenham and Nauta, 1977*). The VTA consists of a large and diverse population of dopamine neurons, as well as smaller populations of purely GABAergic, purely glutamatergic, and GABA/glutamate coreleasing neurons. Both GABAergic and dopaminergic VTA neurons receive input from the LHb (*Beier et al., 2015*; *Lammel et al., 2012*; *Morales and Margolis, 2017*; *Watabe-Uchida et al., 2012*) but it is unknown if these arise from molecularly distinct LHb neurons. We tested if there was connectivity specificity between LHb and VTA neuronal subtypes using rabies virus-based monosynaptic retrograde tracing (*Wickersham et al., 2007*). To examine LHb input to VTA GABAergic neurons we injected Cre-dependent TVA-mCherry into the VTA of a VGAT-IRES-Cre mouse to restrict initial rabies virus infection to GABAergic neurons. We also coinjected a Cre-dependent AAV encoding the rabies glycoprotein (RVG) to allow for retrograde monosynaptic transfer of G-deleted, pseudotyped, rabies virus (EnvA-RbV-GFP) (*Figure 5A*). As only neurons with Cre will express RVG, GFP-labeled neurons in other regions are putatively presynaptic to GFP+/RVG+ VTA neurons (see *Figure 5—figure supplement 2B* for controls for specificity of EnvA-RbV-GFP infection). FISH in the VTA revealed that ~ 30% of 'starter cells' (neurons that were GFP+ and Cre+), coexpressed *Slc17a6* (Vglut2) indicating they are likely GABA/glutamate coreleasing neurons (*Figure 5—figure supplement 2F*) (*Root et al.,*

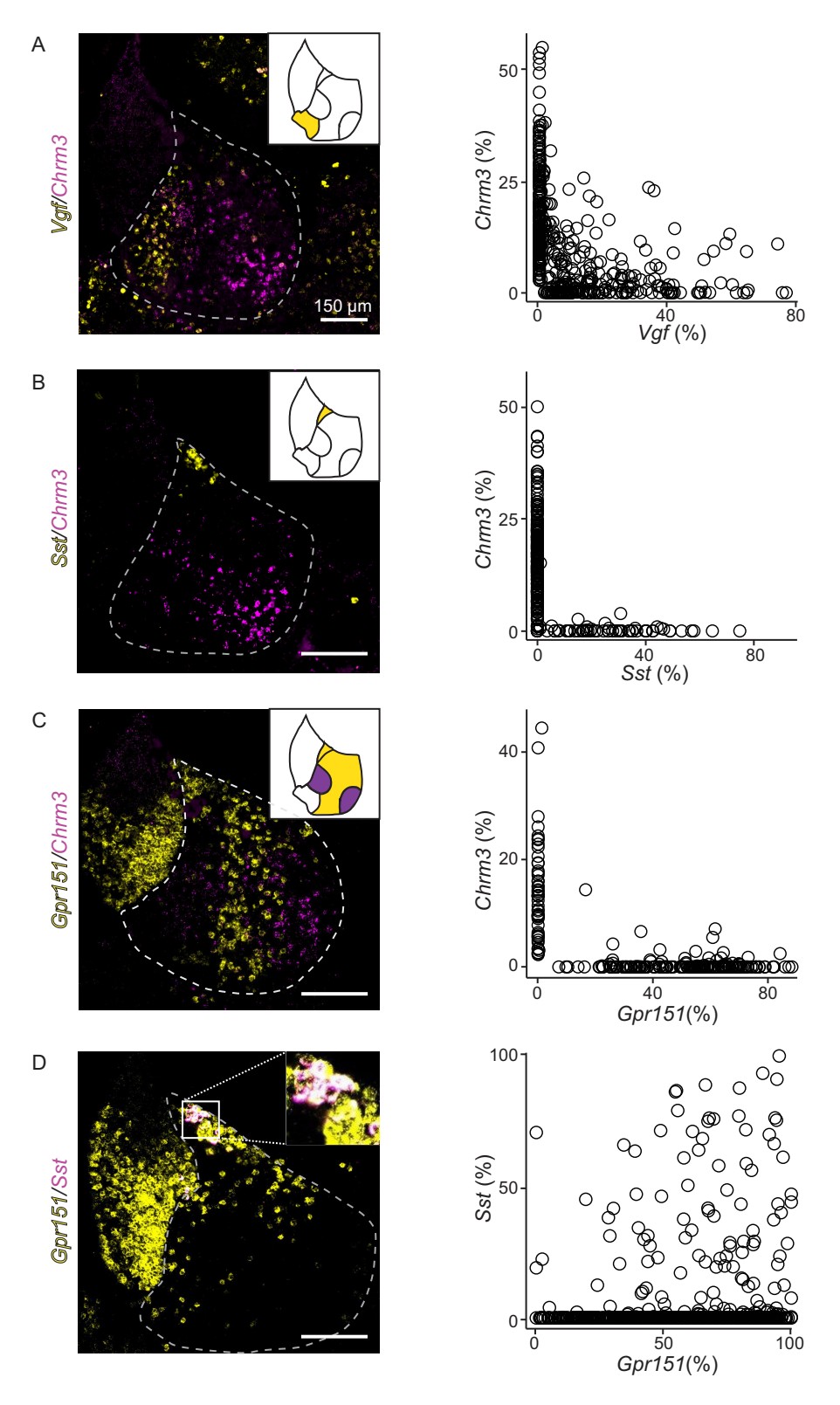

**Figure 4.** FISH confirms that differentially expressed genes from LHb subclusters are nonoverlapping and confined to specific spatial locations of LHb. (**A**) Left: Sample FISH of two differentially expressed LHb genes (*Vgf* (yellow) and *Chrm3* (magenta)), with distinct spatial profiles (LHb outlined with gray dashed line). Right: Quantification of fluorescence coverage of single cells for FISH of *Vgf* and *Chrm3* in LHb neurons (n = 444 cells, three mice). (**B**) Left: Sample FISH of two differentially expressed LHb genes (*Sst* (yellow) and *Chrm3* (magenta)), with distinct spatial profiles. Right: Quantification of

*Figure 4 continued on next page*

*Figure 4 continued*

fluorescence coverage of single cells for FISH of *Sst* and *Chrm3* in LHb neurons (n = 252 cells, three mice). (**C**) Left: Sample FISH of two differentially expressed LHb genes (*Gpr151* (yellow) and *Chrm3* (magenta)), with distinct spatial profiles (illustrated in upper right inset), LHb outlined in gray dashed line. Right: Quantification of fluorescence coverage of single cells for FISH of *Gpr151* and *Chrm3* in LHb neurons (n = 240 cells, three mice). (**D**) Left: Sample FISH of two differentially expressed LHb genes (*Sst* (yellow) and *Gpr151* (magenta)), with similar spatial profiles (both expressed in HbX region). Inset of overlapping *Sst* and *Gpr151* expression in HbX. Right: Quantification of fluorescence coverage of single cells for FISH of *Sst* and *Gpr151* in LHb neurons (n = 112 cells, three mice).

The online version of this article includes the following figure supplement(s) for figure 4:

**Figure supplement 1.** A map of habenula subregions based on single-cell transcriptomic profiling.

**Figure supplement 2.** Spatial distribution of gene expression patterns along the medial/lateral axis for selected LHb DEGs.

**Figure supplement 3.** FISH confirms that differentially expressed genes from the same LHb subcluster are overlapping and confined to similar spatial locations of LHb.

*2014*). The majority of the remaining 70% of 'starter cells' are purely GABAergic (*Figure 5—figure supplement 2F*).

Using FISH we found retrogradely labeled neurons, marked by expression of *RbV-N* mRNA, in all four LHb subtypes (identified using enriched genes *Chrm3, Vgf, Gpr151,* and *Sst*) (*Figure 5B*, *Figure 5—figure supplement 1A*). The majority of retrogradely labeled LHb neurons were found in the lateral and oval/medial subtypes in roughly equal proportions (mean ± SEM: 48 ± 0.5% *Gpr151+* and 41 ± 0.6% *Chrm3+*, respectively) (*Figure 5D*). A much smaller proportion was found in the marginal subtype (10 ± 1% *Vgf+*), and very few HbX neurons (2 ± 1% *Sst+*) were retrogradely labeled. As a population, the distribution of retrogradely labeled cells was skewed laterally along the medial/lateral axis of the LHb (*Figure 5B*).

To examine LHb input to VTA dopaminergic neurons we performed monosynaptic retrograde tracing using the same series of viral injections in DAT-IRES-Cre mice (*Figure 5A*). FISH in the VTA revealed that ~9% of 'starter cells' (neurons that were GFP+ and Cre+), coexpressed *Slc32a1* (VGAT) and *Slc6a3* (dopamine transporter, DAT) indicating they are dopaminergic neurons that also express *Slc32a1* (*Figure 5—figure supplement 2G*) (*Morales and Margolis, 2017*; *Tritsch et al., 2012*). The remaining 91% of starter cells express only *Slc6a3* indicating low levels of starter cell overlap with the experiments done in the VGAT-IRES-Cre line (*Figure 5—figure supplement 2F*). We performed FISH on retrogradely labeled neurons in the LHb and found that a much larger proportion of retrogradely labeled neurons in the oval/medial and marginal subtypes (61 ± 2% *Chrm3+* and 20 ± 1% *Vgf+*, respectively) in the DAT-IRES-Cre than the VGAT-IRES-Cre line (*Figure 5B–D*). Consequently, a smaller proportion of neurons in the lateral subtype were labeled (10 ± 0.4% *Gpr151+*) and almost no neurons in the HbX subregion were labeled (0.7 ± 0.5% *Sst+*) (*Figure 5D*, *Figure 5—figure supplement 1B*). In contrast to the results from injections into VGAT-IRES-Cre mice, the retrogradely labeled cells from injections into DAT-IRES-Cre mice were distributed relatively evenly along the medial/lateral axis of the LHb (*Figure 5C*). Together these data suggest that both VTA GABAergic and dopaminergic neurons can receive input from all four subtypes of LHb neuron. However, VTA dopaminergic neurons receive the largest proportion of their LHb input from the oval/medial LHb subtype, whereas VTA GABAergic neurons receive equal levels of input from both the oval/medial and lateral LHb subtypes (*Supplementary file 6*). Furthermore, the LHb input to VTA GABAergic neurons is skewed laterally in the LHb, while input to dopamine neurons is evenly distributed along the medial/lateral axis (*Figure 5B–C*).

## All LHb neuron subtypes project to the DRN in proportions similar to VTA dopamine neurons

LHb neurons heavily innervate the dorsal and median raphe nuclei (DRN/MRN) and modulate serotonergic output throughout the brain (*Zhao et al., 2015*; *Zhou et al., 2017*). To examine the LHb subtypes that project to the DRN, we injected a non-pseudotyped G-deleted rabies virus (RbV-GFP) into this area and performed FISH in the LHb for subtype enriched genes (*Figure 5—figure supplement 3*). Importantly, non-pseudotyped G-deleted rabies virus does infect axons but does not spread across synapses formed onto the infected cell due to deletion of the rabies glycoprotein (G). Also, as this virus is not pseudotyped, TVA is not needed for infection and the virus acts in a similar way to many other retrograde tracers – but one whose signal we can amplify by performing FISH for

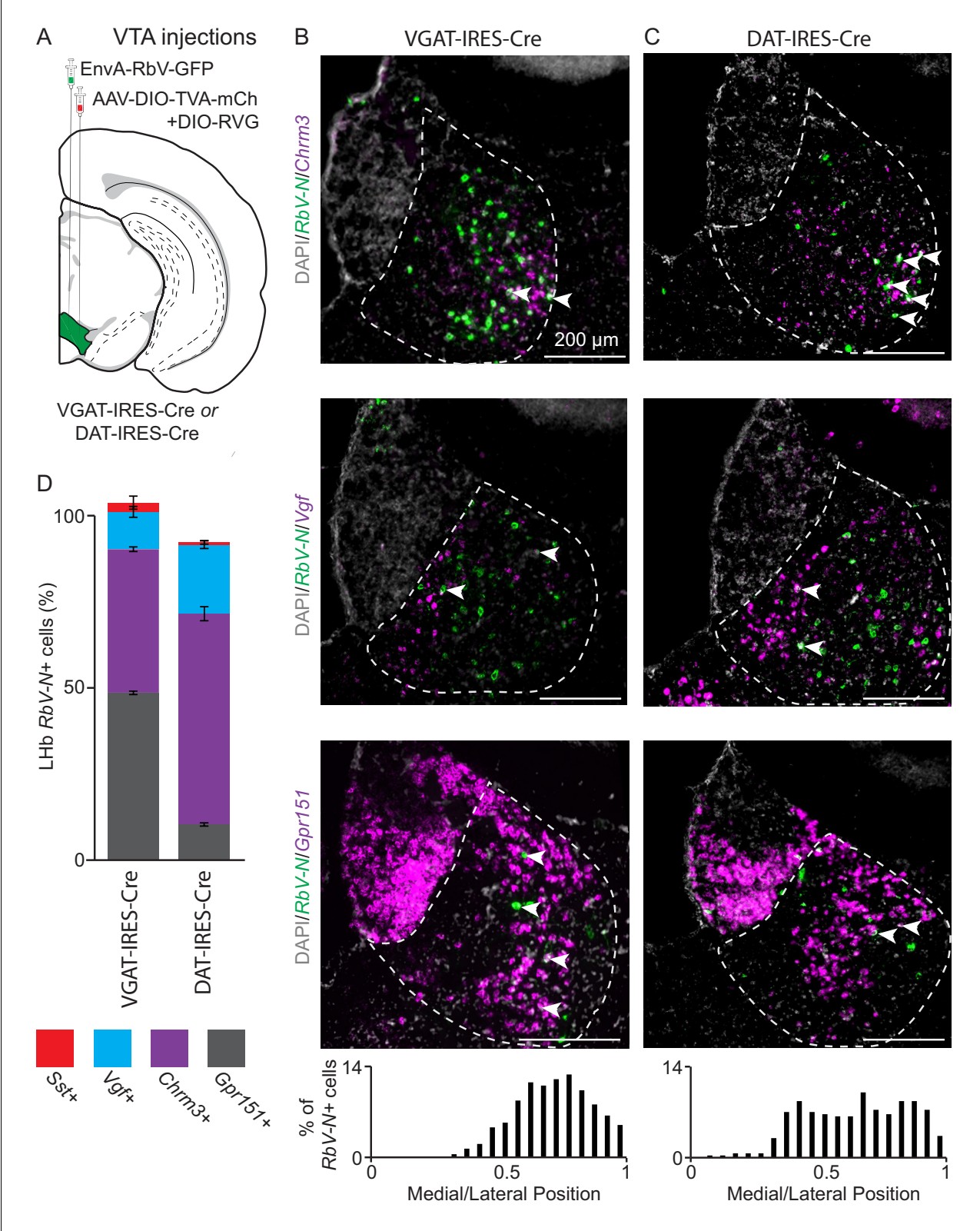

**Figure 5.** Distinct LHb neuron subtypes prefer different downstream targets, but all subtypes target both the VTA. (**A**) Location of site for AAV helper viruses (AAV-FLEX-TVA-mCh and AAV-FLEX-RVG) and pseudotyped rabies virus (EnvA-RbV-GFP) injection into VTA. (**B**) Sample habenula FISH images for *RbV-N* and *Chrm3* (top), *Vgf* (middle), or *Gpr151* (bottom) following viral injection into the VTA of a VGAT-IRES-Cre mouse. Arrow heads show *RbV-N+* cells coexpressing the DEG in each image. Histogram at bottom shows the distribution of the density of *RbV-N+* neurons along the medial lateral

*Figure 5 continued on next page*

*Figure 5 continued*

axis of LHb (n = 822 cells/4 animals). (**C**) Sample habenula FISH images for *RbV-N* and *Chrm3* (top), *Vgf* (middle), or *Gpr151* (bottom) following viral injection into the VTA of a DAT-IRES-Cre mouse. Arrow heads show *RbV-N+* cells coexpressing the DEG in each image. Histogram at bottom shows the distribution of the density of *RbV-N+* neurons along the medial lateral axis of LHb (n = 299 cells/3 animals). (**D**) Quantification of the proportion of *RbV-N* labeled neurons that overlapped with the enriched genes for distinct LHb neuron subtypes (VGAT-IRES-Cre n = 1430 cells/ four mice and DAT-IRES-Cre n = 549/3 mice). Filled rectangles are the mean and error bars are ± SEM, see *Supplementary file 6* for statistical comparisons.

The online version of this article includes the following figure supplement(s) for figure 5:

**Figure supplement 1.** Cells from all four LHb subtypes project to both VGAT-IRES-Cre+ and DAT-IRES-Cre+ cells in the VTA.

**Figure supplement 2.** Quantification and genetic characterization of VTA starter cells from monosynaptic retrograde tracing and controls for rabies virus.

**Figure supplement 3.** Quantification and genetic characterization of LHb cells that project to the DRN using a nonpseudotyped (G-deleted) rabies virus.

the viral gene *RbV-N*. Similar to dopaminergic VTA neurons, the DRN received the largest proportion of its LHb input from the oval/medial subtype (51 ± 2% *Chrm3+*) (***Figure 5—figure supplement 3D***). The DRN also received minor inputs from the lateral (16 ± 5% *Gpr151+*), marginal (28 ± 5% *Vgf +*), and HbX (1 ± 0.5% *Sst+*) regions (***Figure 5—figure supplement 3D***). Overall, the proportions of input to the DRN arising from different LHb subtypes were more similar to those to VTA dopamine neurons than those to VTA GABA neurons (***Figure 5—figure supplement 3D***, ***Supplementary file 6***). However, the distribution of LHb input to DRN has a prominent spike on the medial edge of the LHb indicating that large proportion of the LHb input to DRN is located in this area (***Figure 5—figure supplement 3B and D***). Therefore, the LHb neurons are grossly organized along the medial/lateral axis based on their projection target: the most medial LHb neurons preferentially target the DRN, whereas the most lateral LHb neurons preferentially target VTA GABA neurons. Finally, input to VTA DA neurons is distributed relatively evenly along the medial/lateral axis of the LHb (***Figure 5—figure supplement 3D***).

## Discussion

We performed transcriptional and anatomical analyses of the habenula, a crucial circuit node that modifies brain-wide dopamine and serotonin levels through its connections to the VTA and DRN (***Proulx et al., 2014***; ***Tian and Uchida, 2015***; ***Zhao et al., 2015***). Using large-scale single-cell transcriptional profiling, we classify MHb and LHb neurons into five and four major neuron types, respectively, and show that each class has a distinct gene expression pattern. The four LHb populations were confirmed to be non-overlapping at the single-cell level by FISH. Monosynaptic retrograde tracing revealed that GABAergic VTA neurons receive equal input from the oval/medial and lateral LHb neuronal subtypes, whereas dopaminergic VTA neurons receive input primarily from the oval/medial LHb subtype. Neurons of the DRN receive input from similar LHb cell-types in roughly similar proportions to dopaminergic VTA neurons; however, the LHb cells that target the DRN are skewed medially compared to the cells that target either dopaminergic or GABAergic cells of the VTA.

### Anatomical distribution of MHb neuronal subtypes

Recent studies have identified and delineated the subnuclei of the mouse MHb using morphological, topographic and cytochemical criteria (***Wagner et al., 2016***; ***Wagner et al., 2014***). Using single-cell transcriptional profiling, we show that MHb neurons can be categorized into subtypes based on differential gene expression. Furthermore, the spatial distribution of these transcripts allowed us to ascribe an anatomical location to each subtype. The anatomical location of these subtypes largely agree with previously defined MHb subnuclei and we have used the same nomenclature when possible (***Figure 4—figure supplement 1***) (***Wagner et al., 2016***).

The two ventral subtypes of the MHb coexpressed transcripts for glutamate and ACh neurotransmission (***Figure 2—figure supplement 3***). Our data suggest these two ventral subtypes can be differentially targeted with intersectional approaches, as genes such as *Lmo3* and *Esam* are preferentially expressed in one subtype (***Figure 2D***, ***Figure 2—figure supplement 3***). Previous studies indicate that MHb neurons that release glutamate and ACh target the medial interpeduncular nucleus (IPN) (***Ren et al., 2011***) and are involved in the formation of aversive memories (***Soria-***

*Gómez et al., 2015*). However, whether one or both of the transcriptionally defined subtypes are involved in this process is unknown.

Additionally, cholinergic transmission in MHb has also been implicated in nicotine addiction as MHb neurons not only release ACh, but express an array of nicotinic acetylcholine receptor subunits (nAChRs, such as *Chrna3* and *Chrnb3*; *Figure 2—figure supplements 2* and *6*) (*Fowler et al., 2011*; *Shih et al., 2014*). Similar to its involvement in aversive memories, MHb likely plays an important role in mediating the unpleasant symptoms associated with nicotine withdrawal (*Zhao-Shea et al., 2013*). Our data provide a comprehensive view of all nAChR and mAChR transcripts expressed in both MHb and LHb providing a resource for the development of new therapeutic targets for the treatment of addiction (*Figure 2—figure supplements 6–8*) (*D'Souza, 2016*; *Zuo et al., 2016*).

Few studies have examined the function of the dorsal (enriched for genes *Col16a1*, *Wif1*, and *Adcyap1*) and superior (enriched for genes *Cck*, and *Avil*) MHb. These neurons were known to express high levels of *Tac1* (the gene that produces the neuropeptide substance P), consistent with our single-cell sequencing data (*Figure 2—figure supplement 2J*) and target the lateral IPN (*Hsu et al., 2016*). Their activation may be reinforcing (*Hsu et al., 2014*), but detailed analysis of their function and neurotransmitter release has not been examined.

## LHb neuronal subtypes

We referenced recent studies on LHb subnuclei to create a map (*Figure 4—figure supplement 1*) of LHb based on DEGs extracted from single-cell transcriptional profiling (*Wagner et al., 2016*; *Wagner et al., 2014*). Overall, our map largely agrees with previous work and adds many key observations into the organization and cellular and molecular diversity of the LHb. In addition to providing multiple genetic handles that can be used in future studies to target LHb neuron subtypes, our study reveals the a wide range of GPCRs (such as *Htr2c*, *Htr5b*, *Sstr2*, *Cnr1*, *Gpr158*, *Lpar1*, *Adcyap1r1*; see *Figure 2—figure supplement 8*) expressed in LHb neurons that could be targeted for treatment of diseases known to effect LHb function such as depression, anxiety, and addiction (*Lecca et al., 2014*; *Proulx et al., 2014*). In contrast to some reports (*Zhang et al., 2018*), we did not find evidence of GABAergic neurons in the LHb (or MHb). Although *Gad2* and *Slc6a1*, which encode a GABA synthetic enzyme and GABA transporter, respectively, were present at low levels in all LHb clusters we did not find expression of *Slc32a1* or *Slc18a2*, which are required for vesicular loading of GABA (*Supplementary file 4*). This is in agreement with recently published results demonstrating that *Gad2* expression is a poor discriminator for inhibitory (GABAergic) neurons (*Moffitt et al., 2018*; *Root et al., 2018*). Therefore, either LHb GABAergic cells are rare enough to be missed in the single-cell sequencing, or the habenula is devoid of GABAergic neurons.

We used *Gpr151* expression (as well as *Pbx3* expression) to mark the lateral region of the LHb. The *Gpr151+* neurons of the lateral LHb are the most well-studied neuronal subtype in the LHb and they receive major input from the lateral preoptic area, lateral hypothalamus, entopeduncular nucleus (EP), basal nucleus of the stria terminalis, and the nucleus of the diagonal band (*Broms et al., 2017*). These neurons also receive a minor input from the VTA, and are positioned to receive GABA/glutamate coreleasing input from both the EP and VTA (*Root et al., 2014*; *Wallace et al., 2017*). *Gpr151+* axons, likely arriving from the LHb, heavily innervate the RMTg and central and median raphe nucleus, but not the VTA (*Broms et al., 2015*). Our retrograde-tracing studies from VTA neurons show that *Gpr151+* neurons tend to avoid dopaminergic VTA neurons, but heavily innervated the intermingled GABAergic neurons in this brain region (*Figure 5*). VTA GABAergic interneurons are functionally similar to inhibitory RMTg neurons (both populations inhibit VTA dopaminergic neurons [*Cohen et al., 2012*; *Ji and Shepard, 2007*)]), consistent with our results that both are innervated by lateral LHb. Furthermore, the lateral LHb is likely the major LHb subtype to translate aversive signals to VTA GABAergic neurons which become active following reward omission due to increased LHb input (*Tian and Uchida, 2015*). Overall, these cells are positioned to translate signals arriving from EP to downstream midbrain structures involved in both dopamine and serotonin signaling.

The oval/medial subregion of the LHb expresses high levels of *Chrm3* (as well as *Rbfox1*). Our previous studies indicate that this subtype is positioned to receive purely glutamatergic input from *Pvalb+/Slc16a7+* EP neurons that specifically target the lateral oval nucleus of the LHb (*Wallace et al., 2017*). Additionally, GABA/glutamate coreleasing EP neurons target the lateral oval (as well as the neighboring *Gpr151+* lateral LHb) providing overlapping, but differential EP input to

this subregion (*Wallace et al., 2017*). Electrophysiological analysis has also shown preferential input from EP to the lateral half of LHb, specifically, to the neurons that project to RMTg (*Meye et al., 2016*).

In addition to VTA GABAergic neurons, LHb targets meso-prefrontal VTA dopamine neurons while avoiding other (mesolimbic and nigral) dopamine neurons (*Lammel et al., 2012*). We found that the majority of LHb neurons that projected to VTA dopamine neurons expressed *Chrm3*, a gene enriched in the oval/medial subregion. Furthermore, the distribution of retrogradely labeled LHb neurons that project to VTA dopamine neurons is shifted medially compared to the distribution of those that project to VTA GABA neurons, indicating that VTA dopamine projecting *Chrm3+* LHb neurons reside primarily in the more medial part of the oval/medial subdivision (*Figure 5B–C*). This is consistent with studies that suggest that LHb neurons that target the VTA dopamine neurons may be distinct from those that target the RMTg; therefore, two populations of *Chrm3+* LHb neurons with different synaptic targets may exist (*Lammel et al., 2012*; *Li et al., 2011*; *Maroteaux and Mameli, 2012*). Additional genetic heterogeneity between *Chrm3+* oval and medial subdivisions could be further resolved with higher resolution sequencing methods (*Bakken et al., 2018*; *Tasic et al., 2018*). Nevertheless, our data show that VTA dopamine, and VTA GABAergic neurons are positioned to receive quite different synaptic input from the LHb due to their differential targeting by LHb neuronal subtypes.

The marginal subdivision of the LHb was enriched for *Vgf* and *Plch1* (*Figure 4* and *Figure 4—figure supplements 2–3*). Our retrograde-tracing studies revealed that this subregion projects most heavily to the DRN, and the distribution of LHb input to DRN has a prominent spike on the medial edge of the LHb (*Figure 5—figure supplement 3*). This finding is in agreement to other studies showing labeling of the medial half of the LHb following injections of retrograde tracers into the raphe nucleus (*Lecca et al., 2017*; *Quina et al., 2015*; *Szőnyi et al., 2019*). Interestingly, this region also appears to also receive dense input from serotonergic neurons of the raphe nuclei (*Huang et al., 2019*), and express *Htr2c* as well as several other serotonin receptors (*Figure 2—figure supplement 8*). We expect that this subregion also projects heavily to the lateral dorsal tegmental nucleus (LDTg) and posterior hypothalamic area (PH), as retrograde injections into these areas exclusively label the medial half of the LHb (*Quina et al., 2015*).

## Summary

Progress in defining a function for the habenula has been hindered by incomplete understanding of its constituent cell-types and subregions. This study provides a comprehensive description of the neuronal classes in the lateral and medial habenula based on single-cell transcriptional profiling, FISH, and monosynaptic retrograde tracing (*Figure 4—figure supplement 1*). Future studies will improve our understanding of the function of these habenula cell types by employing current optogenetic, chemogenetic, and electrophysiological approaches for precise control and monitoring of individual habenular populations.

# Materials and methods

**Key resources table**

| Reagent type (species) or resource | Designation | Source or reference | Identifiers | Additional information |
|---|---|---|---|---|
| Strain (*Mus musculus*) | C57Bl/6J | The Jackson Laboratory | Cat# JAX:000664 RRID: IMSR_JAX:000664 | |
| Genetic reagent (*Mus musculus*) | VGAT-IRES-Cre | The Jackson Laboratory | Cat# JAX: 016962 RRID: IMSR_JAX:016962 | |
| Genetic reagent (*Mus musculus*) | DAT-IRES-Cre | The Jackson Laboratory | Cat# JAX: 006660 RRID: IMSR_JAX:006660 | |
| Genetic reagent (non-pseudotyped G-deleted rabies virus) | B19-SADdG-EGFP, RbV-GFP | Other | N/A | Generated in-house (see Materials and methods) $10^9$ IU/mL |

*Continued on next page*

*Continued*

| Reagent type (species) or resource | Designation | Source or reference | Identifiers | Additional information |
|---|---|---|---|---|
| Genetic reagent (pseudotyped G-deleted rabies virus) | EnvA-RbV-GFP | Janelia Viral Tools facility | Addgene# 52487 RRID: Addgene_52487 | $10^8$ IU/mL |
| Commercial assay, kit | RNAscope V1 fluorescent multiplex detection assay reagents | ACDBio | Cat#320851 | |
| Commerical assay, kit | RNAscope V1 fluorescent multiplex detection assay probes | ACDBio | Cat# 456781 Cat# 317321 Cat# 404631 Cat# 556241 Cat# 436381 Cat# 519911 Cat# 437701 Cat# 517421 Cat# 423321 Cat# 319171 Cat# 319191 Cat# 315441 | *V-RABV-gp1* *Gpr151* *Sst* *Plch1* *Pbx3* *Rbfox1* *Chrm3* *Vgf* *Cre* *Slc17a6* *Slc32a1* *Slc6a3* |
| Software, algorithm | inDrops pipeline (Python) | PMID: 26000487 | N/A | https://github.com/indrops/indrops |
| Software, algorithm | R 3.4.4 | R project for statistical computing | RRID:SCR_001905 | https://cran.r-project.org/ |
| Software, algorithm | Seurat 2.3.4 | PMID: 29608179 | RRID:SCR_016341 | https://satijalab.org/seurat/ |
| Software, algorithm | MAST 1.4.1 | PMID: 26653891 | RRID:SCR_016340 | https://bioconductor.org/packages/release/bioc/html/MAST.html |
| Software, algorithm | MATLAB (R2015a) | MathWorks | RRID:SCR_001622 | |
| Software, algorithm | Fiji | PMID: 22743772 | RRID:SCR_002285 | https://imagej.net/Fiji |

## Mice

The following mouse strains/lines were used in this study: C57BL/6J (The Jackson Laboratory, Stock # 000664), VGAT-IRES-Cre (The Jackson Laboratory, Stock # 016962), DAT-IRES-Cre (The Jackson Laboratory, Stock # 006660) (for reference, VGAT gene name is *Slc32a1* and DAT gene name is *Slc6a3*). Animals were kept on a 12:12 regular light/dark cycle under standard housing conditions. All procedures were performed in accordance with protocols approved by the Harvard Standing Committee on Animal Care following guidelines described in the U.S. National Institutes of Health Guide for the Care and Use of Laboratory Animals.

## Adeno-associated viruses (AAVs)

Recombinant AAVs used for retrograde tracing experiments (AAV2/9-CAG-FLEX-TCB-mCherry, AAV2/9-CAG-FLEX-RVG) were commercially obtained from the Boston Children's Hospital Viral Core (Addgene # 48332 and 48333, respectively). Virus aliquots were stored at −80℃, and were injected at a concentration of approximately $10^{11}$ or $10^{12}$ GC/ml, respectively. Controls for the specificity of AAV2/9-CAG-FLEX-TCB-mCherry for cells expressing Cre are included in *Figure 5—figure supplement 2*.

## Rabies viruses

Pseudotyped rabies virus (EnvA-RbV-GFP, SAD B19 strain, Addgene# 52487) was commercially obtained from the Janelia Viral Tools Facility stored at −80℃, and injected at a concentration of approximately $10^8$ IU/ml. Controls showing the requirement of TVA for EnvA-RbV-GFP infection are included in *Figure 5—figure supplement 2*. Non-pseudotyped (G-deleted) rabies viruses used for retrograde tracing (RbV-GFP) were generated in-house (*Wickersham et al., 2010*). Virions were amplified from existing stocks in three rounds of low-MOI passaging through BHK-B19G cells by

transfer of filtered supernatant, with 3 to 4 days between passages. Cells were grown at 35°C and 5% CO$_2$ in DMEM with GlutaMAX (Thermo Scientific, #10569010) supplemented with 5% heat-inactivated FBS (Thermo Scientific #10082147) and antibiotic-antimycotic (Thermo Scientific #15240–062). Virions were concentrated from media from dishes containing virion-generating cells by first collecting and incubating with benzonase nuclease (1:1000, Millipore #70664) at 37°C for 30 min, followed by filtration through a 0.22 μm PES filter. The filtered supernatant was transferred to ultracentrifuge tubes (Beckman Coulter #344058) with 2 ml of 20% sucrose in dPBS cushion and ultracentrifugated at 20,000 rpm (Beckman Coulter SW 32 Ti rotor) at 4°C for 2 hr. The supernatant was discarded and the pellet was resuspended in dPBS for 6 hr on an orbital shaker at 4°C before aliquots were prepared and frozen for long-term storage at −80°C. Nonpseudotyped rabies virus titers were estimated based on a serial dilution method counting infected HEK 293 T cells, and quantified as infectious units per ml (IU/ml).

## Stereotaxic surgeries

Adult mice were anesthetized with isoflurane (5%) and placed in a small animal stereotaxic frame (David Kopf Instruments). After exposing the skull under aseptic conditions, viruses were injected through a pulled glass pipette at a rate of 50 nl/min using an UMP3 microsyringe pump (World Precision Instruments). Pipettes were slowly withdrawn (<100 μm/s) at least 10 min after the end of the infusion. Following wound closure, mice were placed in a cage with a heating pad until their activity was recovered before returning to their home cage. Mice were given pre- and post-operative subcutaneous ketoprofen (10 mg/kg/day) as an analgesic, and monitored daily for at least 4 days post-surgery. Injection coordinates from Bregma for VTA were −3.135 mm A/P, 0.4 mm M/L, and 4.4 mm D/V and for DRN were −6.077 mm A/P, 0.1 mm M/L, and −3.33 mm D/V at −40°. Injection volumes for specific anatomical regions and virus types were as follows VTA: 200 nL AAV (mix of helper viruses), 250 nL EnvA-RbV-GFP (21 days after injection of AAV), DRN: 300 nL of RbV-GFP. Animals injected with rabies virus were perfused 7 days after injection in a biosafety level two animal facility.

## Single-cell dissociation and RNA sequencing

Eight- to 10-week-old C57BL/6J mice were pair-housed in a regular 12:12 light/dark cycle room prior to tissue collection. Mice were transcardially perfused with an ice-cold choline cutting solution (110 mM choline chloride, 25 mM sodium bicarbonate, 12 mM D-glucose, 11.6 mM sodium L-ascorbate, 10 mM HEPES, 7.5 mM magnesium chloride, 3.1 mM sodium pyruvate, 2.5 mM potassium chloride, 1.25 mM sodium phosphate monobasic, saturated with bubbling 95% oxygen/5% carbon dioxide, pH adjusted to 7.4 using sodium hydroxide). Brains were rapidly dissected out and sliced into 200-μm-thick coronal sections on a vibratome (Leica Biosystems, VT1000) with a chilled cutting chamber filled with choline cutting solution. Coronal slices containing the habenula were then transferred to a chilled dissection dish containing a choline-based cutting solution for microdissection. Dissected tissue chunks were transferred to cold HBSS-based dissociation media (Thermo Fisher Scientific Cat. # 14170112, supplemented to final content concentrations: 138 mM sodium chloride, 11 mM D-glucose, 10 mM HEPES, 5.33 mM potassium chloride, 4.17 mM sodium bicarbonate, 2.12 mM magnesium chloride, 0.441 mM potassium phosphate monobasic, 0.338 mM sodium phosphate monobasic, saturated with bubbling 95% oxygen/5% carbon dioxide, pH adjusted to 7.35 using sodium hydroxide) and kept on ice until dissections were completed. Dissected tissue chunks for each sample were pooled for each hemisphere for the subsequent dissociation steps. Tissue chunks were first mixed with a digestion cocktail (dissociation media, supplemented to working concentrations: 20 U/ml papain, 0.05 mg/mL DNAse I) and incubated at 34°C for 90 min with gentle rocking. The digestion was quenched by adding dissociation media supplemented with 0.2% BSA and 10 mg/ml ovomucoid inhibitor (Worthington Cat. # LK003128), and samples were kept chilled for the rest of the dissociation procedure. Digested tissue was collected by brief centrifugation (5 min, 300 g), re-suspended in dissociation media supplemented with 0.2% BSA, 1 mg/ml ovomucoid inhibitor, and 0.05 mg/mL DNAse I. Tissue chunks were then mechanically triturated using fine-tip plastic micropipette tips of progressively decreasing size. The triturated cell suspension was filtered through a 40 μm cell strainer (Corning 352340) and washed in two repeated centrifugation (5 min, 300 g) and re-suspension steps to remove debris before a final re-suspension in dissociation media containing 0.04% BSA and 15% OptiPrep (Sigma D1556). Cell density was calculated based on hemocytometer

counts and adjusted to approximately 100,000 cells/ml. Single-cell encapsulation and RNA capture on the InDrop platform was performed at the Harvard Medical School ICCB Single Cell Core using v3 chemistry hydrogels based on previously described protocols (*Zilionis et al., 2017*). Suspensions were kept chilled until the cells were flowed into the microfluidic device. Libraries were prepared and indexed following the protocols referenced above, and sequencing-ready libraries were stored at −80℃. Libraries from different samples were pooled and sequenced on an Illumina NextSeq 500 (High Output v2 kits).

## Sequencing data processing

NGS data was processed using previously a published pipeline in Python available at [https://github.com/indrops/indrops] (*Klein et al., 2015*). Briefly, reads were filtered by expected structure and sorted by the corresponding library index. Valid reads were then demultiplexed and sorted by cell barcodes. Cell barcodes containing fewer than 250 total reads were discarded, and remaining reads were aligned to a reference mouse transcriptome (Ensembl GRCm38 release 87) using Bowtie 1.1.1 (m = 200, n = 1, l = 15, e = 1000). Aligned reads were then quantified as UMI-filtered mapped read (UMIFM) counts. UMIFM counts and quantification metrics for each cell were combined into a single file sorted by library and exported as a gunzipped TSV file.

## Pre-clustering filtering and normalization

Analysis of the processed NGS data was performed in R (version 3.4.4) using the Seurat package (version 2.3.4) (*Butler et al., 2018*; *Satija et al., 2015*). Cells with fewer than 500 UMIFM counts and 200 genes were removed. The expression data matrix (Genes x Cells) was filtered to retain genes with >5 UMIFM counts, and then loaded into a Seurat object along with the library metadata for downstream processing. The percentage of mitochondrial transcripts for each cell (percent.mito) was calculated and added as metadata to the Seurat object. Cells in the object were further filtered using the following parameters: nUMI – min. 500, max. 18000; nGene – min. 200, max. 6000; percent.mito – min. -Inf, max. 0.1. Low-quality libraries identified as outliers on scatter plots of quality control metrics (e.g. unusually low gradient on the nGene vs. nUMI) were also removed from the dataset. Filtered Seurat objects were then log-normalized at 10,000 transcripts per cell. Effects of latent variables (nUMI, percent.mito) were estimated and regressed out using a GLM (`ScaleData` function, `model.use` = 'linear'), and the scaled and centered residuals were used for dimensionality reduction and clustering. Raw data and the normalized, filtered, scaled, R object can be found at Harvard Dataverse here: https://doi.org/10.7910/DVN/2VFWF6.

## Cell clustering and cluster identification

Initial clustering was performed on the dataset using the first 20 PCs, and t-SNE was used only for data visualization. Clustering was run using the SNN-based `FindClusters` function using the SLM algorithm and 10 iterations. Clustering was performed at varying resolution values, and we chose a final value of 1.2 for the resolution parameter for this stage of clustering. Clusters were assigned preliminary identities based on expression of combinations of known enriched genes for major cell classes and types. The full list of enriched genes is provided in *Supplementary file 2* and average expression of all genes in all clusters is provided in *Supplementary file 1*. Low quality cells were identified based on a combination of low gene/UMIFM counts and high levels of mitochondrial and nuclear transcripts (e.g. *Malat1*, *Meg3*, *Kcnq1ot1*) typically clustered together and were removed. Following assignment of preliminary identities, cells were divided into data subsets as separate Seurat objects (LHb neurons and MHb neurons) for further subclustering. The expression matrix for each data subset was further filtered to include only genes expressed by the cells in the subset (minimum cell threshold of 0.5% of cells in the subset). Subclustering was performed iteratively on each data subset to resolve additional cell types and subtypes. Briefly, clustering was run at high resolution, and the resulting clusters were ordered in a cluster dendrogram using the `BuildClusterTree` function in Seurat which uses cluster averaged PCs for calculating a PC distance matrix. Putative doublets/multiplets were identified based on expression of known enriched genes for different cell types not in the cell subset (e.g. neuronal and glial-specific genes). Putative doublets tended to separate from other cells and cluster together, and these clusters were removed from the dataset. Cluster separation was evaluated using the `AssessNodes` function and inspection of differentially

expressed genes at each node. Clusters with poor separation, based differential expression of mostly housekeeping genes, or activity dependent genes (see *Figure 2—figure supplement 1*) were merged to avoid over-separation of the data. The dendrogram was reconstructed after merging or removal of clusters, and the process of inspecting and merging or removing clusters was repeated until all resulting clusters could be distinguished based on a set of differentially expressed genes that we could validate separately. To calculate the 'ADG Score' (*Figure 2—figure supplement 1*), we used the `AddModuleScore` function in Seurat using a list of ADGs that were highly expressed in some of the MHb clusters (*Fos, Fosb, Egr1, Junb, Nr4a1, Dusp18, Jun, Jund*).

## Differential expression tests

Tests for differential gene expression were performed using MAST (version 1.10.1) (*Finak et al., 2015*) through the `FindMarkersNode` function in Seurat (`logfc.threshold = 0.25, min. pct = 0.1`). Adjusted *P* values were corrected using the Bonferroni correction for multiple comparisons ($p < 0.05$).

## Fluorescence in-situ hybridization (FISH)

Mice were deeply anesthetized with isoflurane, decapitated, and their brains were quickly removed and frozen in tissue freezing medium on dry ice. Brains were cut on a cryostat (Leica CM 1950) into 30 μm sections, adhered to SuperFrost Plus slides (VWR), and immediately refrozen. Samples were fixed 4% paraformaldehyde and processed according to ACD RNAscope Fluorescent Multiplex Assay manual. Sections were incubated at room temperature for 30 s with DAPI, excess liquid was removed, and immediately coverslipped with ProLong antifade reagent (Molecular Probes). Antisense probes for *RbV-N, Gpr151, Sst, Plch1, Pbx3, Rbfox1, Chrm3, Vgf, Cre, Slc17a6, Slc32a1, and Slc6a3* were purchased from Advanced Cell Diagnostics (ACD, http://acdbio.com/). Sections were imaged at 1920 × 1440 pixels on a Keyence BZ-X710 fluorescence microscope using a 10X, 0.45 NA air Nikon Plan Apo objective. Individual imaging planes were overlaid and quantified for colocalization in ImageJ (NIH) and Matlab (Mathworks).

## Image analysis

FISH images were analyzed for 'fluorescence coverage (%)," meaning the proportion of fluorescent pixels to total pixels in a cellular ROI, using a custom macro in Image J and custom scripts in Matlab (*Figure 4*, *Figure 4—figure supplements 2–3*, and *Figure 5—figure supplements 1–3*). 5–10 images from at least three mice were analyzed for each condition. Cell ROIs were automatically determined based on fluorescence signals in all three channels (or by fluorescence in the *RbV-N* channel for rabies tracing experiments), and manually adjusted prior to analysis to ensure that all cell ROIs reflected individual cells and not clusters. After background subtraction (the signal outside of cell ROIs) and application of a fluorescence threshold (Renyi Entropy), the amount of fluorescent pixels in each optical channel was counted within the cellular ROI. All images compared underwent identical thresholding and no other manipulations were made. These data were used to generate X-Y plots displaying the percent coverage for each channel per cell (*Figure 4*, *Figure 4—figure supplements 2–3*, and *Figure 5—figure supplements 1–3*). For all histograms of media/lateral position of cells (*Figure 4—figure supplement 2*, *Figure 5*, and *Figure 5—figure supplement 3*), a subset of images were chosen that excluded the anterior and posterior poles of the habenula (images included spanned approximately −1.655 to −1.855 A/P from bregma). The anterior and posterior poles of the LHb were excluded for this analysis because the subregions described in *Figure 4—figure supplement 1* did not persist. Images including the anterior and posterior poles of the habenula were included for all other analysis except for these histograms.

## Acknowledgements

The authors thank Sarah Melzer and Adam Granger for assistance with FISH analysis; James Levasseur for animal husbandry and genotyping; L Worth for administrative assistance; HMS ICCB Single Cell Core for assistance with scRNA-seq experiments on the InDrop platform; The Bauer Core Facility at Harvard University for sequencing support and the members of the Sabatini Lab for their helpful discussions and advice. Starting materials for generating nonpseudotyped rabies virus is a generous gift from BK Lim (UCSD). This work was supported by the Howard Hughes Medical

Institute (BLS), NIH, National Institute of Neurological Disease and Stroke (K99 NS105883 to MLW and NS103226 to BLS).

## Additional information

### Funding

| Funder | Grant reference number | Author |
|---|---|---|
| Howard Hughes Medical Institute | | Bernardo L Sabatini |
| National Institute of Neurological Disorders and Stroke | NS103226 | Bernardo L Sabatini |
| National Institute of Neurological Disorders and Stroke | NS105883 | Michael Wallace |

The funders had no role in study design, data collection and interpretation, or the decision to submit the work for publication.

### Author contributions

Michael L Wallace, Conceptualization, Resources, Data curation, Formal analysis, Investigation, Methodology, Project administration; Kee Wui Huang, Software, Investigation; Daniel Hochbaum, Data curation, Formal analysis, Investigation, Methodology; Minsuk Hyun, Gianna Radeljic, Data curation, Investigation, Methodology; Bernardo L Sabatini, Conceptualization, Resources, Supervision, Project administration

### Author ORCIDs

Michael L Wallace (ID) https://orcid.org/0000-0002-7270-8521
Kee Wui Huang (ID) http://orcid.org/0000-0003-2265-4550
Bernardo L Sabatini (ID) https://orcid.org/0000-0003-0095-9177

### Ethics

Animal experimentation: All procedures were performed in accordance with protocols approved by the Harvard Standing Committee on Animal Care following guidelines described in the US National Institutes of Health Guide for the Care and Use of Laboratory Animals (HMS IACUC protocol #IS00000571). All surgery was performed under isoflurane anesthesia.

### Decision letter and Author response

Decision letter https://doi.org/10.7554/eLife.51271.sa1
Author response https://doi.org/10.7554/eLife.51271.sa2

## Additional files

### Supplementary files

• Supplementary file 1. Average expression of all genes in all clusters.

• Supplementary file 2. Differentially expressed genes from all clusters using MAST, dendrogram included for reference.

• Supplementary file 3. Differentially expressed genes from MHb subclusters using MAST, dendrogram included for reference.

• Supplementary file 4. Average expression of all genes from MHb and LHb subclusters.

• Supplementary file 5. Differentially expressed genes from LHb subclusters using MAST, dendrogram included for reference.

• Supplementary file 6. Statistical comparisons for retrograde labeling experiments shown in *Figure 5* and *Figure 5—figure supplement 3*.

- Transparent reporting form

## Data availability

Sequencing data have been deposited in harvard dataverse at the following link: https://doi.org/10.7910/DVN/2VFWF6.

The following dataset was generated:

| Author(s) | Year | Dataset title | Dataset URL | Database and Identifier |
|-----------|------|---------------|-------------|-------------------------|
| Michael Wallace | 2019 | Wallace_etal_2019_habenula_scseq | https://doi.org/10.7910/DVN/2VFWF6 | Harvard Dataverse, 10.7910/DVN/2VFWF6 |

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
