## [Decision Letter]

**Acceptance summary:**

This study provides a new and detailed molecular characterization of medial and lateral habenula cell subtypes. This analysis occurs at the anatomical and projection targeting levels using an array of modern approaches including single cell sequencing, FISH, and mononsynaptic tracing. The authors provide new information into molecular identities of lateral and medial habenula cell subtypes that are organized into anatomical subregions. They further show that lateral habenula anatomical neuron populations display distinct targeting to dopaminergic and GABAergic neurons in the ventral tegmental area. This is a timely study that provides a valuable resource for researchers studying habenula cell subtypes.

**Decision letter after peer review:**

Thank you for submitting your article "Distinct neuronal subtypes of the lateral habenula differentially target ventral tegmental area dopamine neurons" for consideration by *eLife*. Your article has been reviewed by three peer reviewers, including Mary Kay Lobo as the Reviewing Editor and Reviewer #1, and the evaluation has been overseen by Kate Wassum as the Senior Editor. The following individual involved in review of your submission has agreed to reveal their identity: Manuel Mameli (Reviewer #3).

The reviewers have discussed the reviews with one another and the Reviewing Editor has drafted this decision to help you prepare a revised submission.

Essential revisions:

The reviewers raise a number of concerns that must be adequately addressed before the paper can be accepted. Some of the required revisions will likely require further experimentation within the framework of the presented studies and techniques.

1) Please clarify how the DEG mapping to anatomy was performed. It is unclear if the mapping relates to one of the DEGs or many. Analysis such as ISH patterns of top DEGs for each molecular subgroup could be overlapped to create a density map to provide a better sense of how the DEGs that strongly drove clustering map to location.

2) Please provide information on the total number of starter cells that were labeled with the RABV within the VTA and DRN including images in the figures. If this represents a small proportion, it is possible that only a fraction of the total LHb neurons were retrogradely labeled. Please also provide data showing the specificity to dopamine or GABA starter neurons (i.e. expand Figure 5—figure supplement 2).

3) Please clarify in the Results, Materials and methods, and figures the vector used for the retrograde labeling in the DRN. If G-containing RABV was used the virus could jump more than a synapse over time, rendering the interpretation difficult.

4) The data in Figure 5 could be improved to demonstrate the differential targeting of VTA neuron subtypes and the DRN. It is difficult to visualize *Chrm3* and *RbV-N* colocalization. Enlarged images similar to Figure 4 are also needed to improve this visualization. Additionally, please add images for *Gpr151, Vgf*, and *Sst/Gpr151*.

5) The experiments performed in the DRN lack the detailed analysis performed in the VTA. The DRN and MRN contain different neuronal populations, and the LHb seems to innervate serotonin containing neurons (Lecca et al., 2017). If the authors intend to keep the DRN data in this study, then please provide a more refined connectivity analysis similar to the VTA analysis. Please also provide discussion on the DRN data.

6) An issue in the LHb field relates to the medial-lateral subdivisions and whether the structure is better thought of in terms of compartments or gradients. It seems the authors are in a good position to add to this debate with their data but elect to "fit" their data into existing structures while noting the gradients in the text. For example, *Chrm3* staining in Figure 4B seems to cover the entire ventral portion. Please include descriptions of gradient expression as well. Additional, ISHs of these targets themselves or somehow obtaining this data from the Allen Brain could address this issue.

7) In the second paragraph of the subsection “Anatomical distribution of MHb neuronal subtypes”, it is difficult to tell from mean z-scored data and the circle plots that genes such as *Lm03* and *Esam* are preferentially expressed in one subtype. Please address this.

---

## [Author Response]

Essential revisions:The reviewers raise a number of concerns that must be adequately addressed before the paper can be accepted. Some of the required revisions will likely require further experimentation within the framework of the presented studies and techniques.1) Please clarify how the DEG mapping to anatomy was performed. It is unclear if the mapping relates to one of the DEGs or many. Analysis such as ISH patterns of top DEGs for each molecular subgroup could be overlapped to create a density map to provide a better sense of how the DEGs that strongly drove clustering map to location.

In order to determine the spatial location of each genetically defined LHb subgroup we have now analyzed gene expression patterns using FISH for two DEGs for each subgroup (Figure 4, Figure 4—figure supplement 2 and Figure 4—figure supplement 3). Using FISH we show that expression of DEGs from the same subgroup overlap (Figure 4—figure supplement 3), while expression of DEGs from different subgroups is mutually exclusive (Figure 4). We also show that the spatial distribution along the medial/lateral axis of each of these DEGs is distinct between subgroups, but similar within a subgroup (Figure 4—figure supplement 2). Using the Allen Brain Atlas, (Lein et al., 2007) we have also examined the expression of many other LHb and MHb DEGs and find that they show similar expression patterns to the genes that for which we performed FISH analysis. Therefore, we are confident that the expression patterns of the DEGs we chose to highlight with FISH (i.e. *Sst, Chrm3, Gpr151, Vgf, Rbfox1, Pbx3,* and *Plch1*) are representative of the gene expression patterns of each LHb subgroup. The illustration in Figure 4—figure supplement 1 summarizes these observations. Please also see response to Essential revision 6 below.

2) Please provide information on the total number of starter cells that were labeled with the RABV within the VTA and DRN including images in the figures. If this represents a small proportion, it is possible that only a fraction of the total LHb neurons were retrogradely labeled.

Approximately 12% (451/3754 cells/4 mice) of VGAT+ VTA neurons were starter cells in VGAT-IRES-Cre retrograde labeling experiments. In DAT-IRES-Cre experiments, approximately 18% (468/2598 cells/3 mice) of DAT+ VTA neurons were starter cells (Figure 5—figure supplement 2). Therefore, as in all studies using psuedotyped G-deleted rabies virus, a small subset of the total population of VTA neurons were putative starter cells. These data are now included in the legend for Figure 5—figure supplement 2.

Unlike the experiments in the VTA, the DRN experiments used a nonpsuedotyped G-deleted rabies virus (referred to in figures as RbV-GFP); therefore, in these experiments there were no starter cells (Figure 5—figure supplement 3). The nonpseudotyped G-deleted rabies virus is axon infecting, but does not spread across synapses due to deletion of the rabies glycoprotein (G). As the virus is nonpseudotyped, TVA is no longer needed for infection and the virus acts in a similar way to many other retrograde tracers such as cholera toxin or retrobeads, but we can amplify its signal by performing FISH for the viral gene *RbV-N*.

Please also provide data showing the specificity to dopamine or GABA starter neurons (i.e. Expand Figure 5—figure supplement 2).

To examine the specificity of our starter cell population in the VGAT-IRES-Cre and DAT-IRES-Cre monosynaptic tracing experiments we injected AAV-DIO-TVA, but not AAV-DIO-RVG, followed by EnvA-RbV-GFP into the VTA. This allows initial infection of pseudotyped rabies virus into cells expressing the TVA receptor, but due to omission of RVG, does not allow subsequent spread of the virus. Therefore, all cells expressing *RbV-N,* but not *Cre* are due to non-specific expression of TVA. Our new data show that 99% of *RbV-N+* cellsin the VGAT-IRES-Cre and 95% *RbV-N+* cells in the DAT-IRES-Cre mice also express *Cre* indicating high specificity of our viruses for *Cre+* cells in VTA. Of course, these percentages are likely underestimates due to false negatives (i.e. inability to detect Cre by FISH in Cre-expressing neurons). These data are now included in Figure 5—figure supplement 2C.

3) Please clarify in the Results, Materials and methods, and figures the vector used for the retrograde labeling in the DRN. If G-containing RABV was used the virus could jump more than a synapse over time, rendering the interpretation difficult.

As stated above (see Essential revision2), the DRN experiments used a nonpsuedotyped G-deleted rabies virus (Figure 5—figure supplement 3). The nonpseudotyped G-deleted rabies virus is axon infecting, but does not spread across synapses due to deletion of the rabies glycoprotein (G). As the virus is nonpseudotyped, TVA is no longer needed for infection and the virus acts in a similar way to many other retrograde tracers such as cholera toxin or retrobeads. We have clarified this detail in the Materials and methods, Results, and figure legends. Additionally, we have moved all of the DRN data to supplementary figures (Figure 5—figure supplement 3) to diminish confusion due to differences in methodology in the two types of experiments.

4) The data in Figure 5 could be improved to demonstrate the differential targeting of VTA neuron subtypes and the DRN. It is difficult to visualize Chrm3 and RbV-N colocalization. Enlarged images similar to Figure 4 are also needed to improve this visualization. Additionally, please add images for Gpr151, Vgf, and Sst/Gpr151.

We added larger images to visualize *RbV-N* colocalization with DEGs in Figure 5 and Figure 5—figure supplement 3. We also included histograms depicting the medial/lateral distribution of *RbV-N*+ cells for each injection location/cell-type. Interestingly, these new analyses revealed different distributions of retrogradely labeled cells for each target region/cell-type.

5) The experiments performed in the DRN lack the detailed analysis performed in the VTA. The DRN and MRN contain different neuronal populations, and the LHb seems to innervate serotonin containing neurons (Lecca et al., 2017). If the authors intend to keep the DRN data in this study, then please provide a more refined connectivity analysis similar to the VTA analysis. Please also provide discussion on the DRN data.

All of the DRN data has been moved to Figure 5—figure supplement 3. This experiment used a non-pseudotyped rabies (G-deleted) virus (referred to in the figures as RbV-GFP) which differs from the pseudotyped rabies (G-deleted) virus (referred to in figures as EnvA-RbV-GFP) used in the VGAT-IRES-Cre and DAT-IRES-Cre experiments (see Essential revisions 2 and 3). Therefore, due to differences in methodology, it is more appropriate to present these data in separate figures. We believe that this showing this data is worthwhile as it despite the fact that there is no genetically defined starter cell population. We also include discussion of this data and new analysis in the Results and Discussion section.

6) An issue in the LHb field relates to the medial-lateral subdivisions and whether the structure is better thought of in terms of compartments or gradients. It seems the authors are in a good position to add to this debate with their data but elect to "fit" their data into existing structures while noting the gradients in the text. For example, Chrm3 staining in Figure 4B seems to cover the entire ventral portion. Please include descriptions of gradient expression as well. Additional, ISHs of these targets themselves or somehow obtaining this data from the Allen Brain could address this issue.

We replaced the image in Figure 4B with an image that is more representative of pattern of *Chrm3* we observe. Also, we now show the distribution of cells positive for DEGs *Chrm3, Gpr151,* and *Vgf* as well as expression levels in individual cells along the medial/lateral axis of LHb in Figure 4—figure supplement 2.

To show that these patterns of DEG expression are representative of each cluster rather than unique to those particular DEGs, we performed FISH with a new set of DEGs: *Rbfox1, Pbx3,* and *Plch1*, and analyzed gene expression and location. These new genes were chosen because, similar to *Chrm3, Gpr151,* and *Vgf,* they are DEGs that are enriched in a specific LHb cluster (Oval/medial, lateral, and marginal, respectively). In Figure 4—figure supplement 3, using FISH we show that *Rbfox1, Pbx3,* and *Plch1* are expressed by the same cells as *Chrm3, Gpr151,* and *Vgf,* respectively. In Figure 4—figure supplement 2, we show that the distribution of cells positive for *Rbfox1, Pbx3,* and *Plch1* along the medial/lateral axis also matches the distribution of *Chrm3, Gpr151,* and *Vgf*. We did not observe obvious gradients in expression along the medial/lateral axis for most of the genes; however, *Plch1* expression did gradually decrease in expression for cells positioned more lateral within the LHb. These new results and analysis are now described and discussed in the Results and Discussion sections.

7) In the second paragraph of the subsection “Anatomical distribution of MHb neuronal subtypes”, it is difficult to tell from mean z-scored data and the circle plots that genes such as Lm03 and Esam are preferentially expressed in one subtype. Please address this.

In Figure 2—figure supplement 3, we have added violin plots showing gene expression levels (log normalized and scaled) of DEGs for all MHb and LHb clusters shown in Figure 2 and 3. These plots show the distribution of expression levels for *Lmo3* and *Esam* for all clusters, illustrating their preferential expression in the ventral 2/3 of MHb, and the ventrolateral MHb clusters, respectively.